# Lightning activity in Northern Europe during a stormy winter: disruptions of weather patterns originating in global climate phenomena

Ivana Kolmašová[1,2], Ondřej Santolík[1,2], Kateřina Rosická[2]

[1] Department of Space Physics, Institute of Atmospheric Physics, Czech. Acad. Sci., Prague, 141 00, Czechia
[2] Faculty of Mathematics and Physics, Charles University, Prague, 121 16, Czechia

*Correspondence to*: Ivana Kolmašová (iko@ufa.cas.cz)

**Abstract.** In this study, we use the World Wide Lightning Location Network data and investigate properties of more than ninety thousand lightning strokes which hit Northern Europe during an unusually stormy winter 2014/2015. Thunderstorm
days with at least two strokes hitting an area of 0.5° x 0.5° occurred 5-13 times per month in the stormiest regions. Such frequency of thunderstorm days is about five times higher than a mean annual number calculated for the same region over winter months in 2008-2017. The number of individual winter lightning strokes was about four times larger than the long-term median calculated over the last decade. In colder months of December, January and February, the mean energy of detected strokes was by two order of magnitude larger than the global mean stroke energy of 1 kJ. We show for the first time that winter
superbolts with radiated electromagnetic energies above one mega joule appeared at night and in the morning hours, while the diurnal distribution of all detected lightning was nearly uniform. We also show that the superbolts were often single stroke flashes and that their subsequent strokes never reached MJ energies. The lightning strokes were concentrated above the ocean close to the western coastal areas. All these lightning characteristics suppose an anomalously efficient winter thundercloud charging in the eastern North Atlantic, especially at the sea-land boundary. We found that the resulting unusual production of
lightning could not be explained solely by an anomalously warm sea surface caused by a positive phase of the North Atlantic Oscillation and by a starting super El Nino event. Increased updraft strengths, which are believed to accompany the cold to warm transition phase of El Nino, might have acted as another charging driver. We speculate that a combination of both these large-scale climatic events might have been needed to produce observed enormous amount of winter lightning in winter 2014/2015.

## 1 Introduction

Thunderstorms, which occur during winter months, are often accompanied by very strong gusty winds, heavy precipitation in a form of snow, rain or hail, and occasionally by very energetic lightning (Schultz & Vavrek, 2009). A necessary meteorological condition for generation of wintertime thunderclouds is the spread of a cold air over a warmer lake, ocean or seawater (Williams, 2018) which can result in an ascent of the air warmer than its surroundings. Therefore, there are only three regions at the northern hemisphere where the winter storms regularly produce numerous lightning flashes: Japan, Mediterranean and the USA (Montanyà et al., 2016). Taszarek et al. (2019) investigated the climatology of thunderstorm days (TDs) in Europe using the data from the European lightning detection network EUCLID (European Cooperation for Lightning Detection) for the period of 2008-2017 (their Fig. 4). To determine TDs, the area of Europe was divided in 0.5° x 0.5° bins. A TD was defined for each bin as a day with at least two detected strokes. They found that during winter seasons there were on average 3-7 TDs in individual bins per month in Mediterranean but only 1-2 TDs per month in the Northern Europe and in British Islands. That is why an occurrence of winter lightning in these regions is rare and can attract the attention of scientists and journalists. This happened at the beginning of the twentieth century, when winter lightning unexpectedly occurred above the British Islands. The British Meteorological Office asked readers of the Nature journal to assist in investigation of winter thunderstorms by sending postcards with reports of the time, position and number of observed lightning flashes (Cave, 1916; 1923). As a result, monthly amounts of flashes eye witnessed by the Nature readers living in different parts of the British Islands were published (Bower, 1926; 1927).

Today, we do not need to relay on eyewitnesses and their postcards. Lightning strokes excite electromagnetic pulses, which are now routinely used to localize lightning by triangulation techniques based on networks of radio receivers, and data provided by lightning location networks are ordinarily used for different lightning studies. Montanyà et al. (2016) used the data from the World Wide Lightning Location Network (WWLLN; Rodger et al., 2004) and provided for the first time world maps with winter lightning activity. Adhikari and Liu (2019) searched for thundersnow events in WWLLN data from 2010-2015 in regions with very low surface temperatures. Ninety percent of snow lightning events were found to occur over high mountainous regions. Low-elevation thundersnow events were observed exclusively above land. The thundersnow events occurred more frequently in evening and pre-midnight hours. Holzworth et al. (2019) examined the WWLLN data from 2010 to 2018 focusing on superbolts with stroke energies above 1 MJ, it means with energies by three orders of magnitude larger than the mean energy of all lightning strokes detected by WWLLN. The distribution of superbolts globally peaked in the Northern Hemisphere winter from November to February in the European North Atlantic region and in the Mediterranean, and appeared predominantly over water. Similarly, Turman (1977) analysed satellite-based optical measurements and found a majority of superbolts (defined here as strokes with the optical power above $3 \times 10^{12}$ W) located over the oceans or in coastal areas. Numerous studies in the past have shown that the intensity of lightning strokes over the oceans is greater than over the land (Fullekrug et al., 2002; Light et al., 2003; Said et al., 2013; Zoghzoghy et al. 2015). An absence of a clear explanation of

this phenomenon raises a question if the land/ocean lightning intensity contrast is natural or if it can be due to different methods of detection and their efficiency, their frequency band (ELF, VLF, VHF), or propagation effects of these radio waves. Nag and Cummins (2016) analysed differences in the velocity of negative first stroke leaders occurring during the oceanic and land thunderstorms. They hypothesized that the observed difference originates in different thundercloud charge structures, when oceanic thunderclouds lacks a turbulent mixing of cloud hydrometeors which results in more extensive main negative charge regions necessary to produce energetic lightning strokes. The increase of the lightning cadence at the French Atlantic coastline was reported by Seity et al. (2001), and explained by a sudden vertical development of the thundercloud at the coastline allowing more efficient cloud charging. Asfur et al. (2020) recently performed laboratory experiments, which showed that intensity of discharges increased exponentially with the concentration of dissolved salts in the water. Chronis et al. (2016) examined the temporal and spatial variations (2004–2010) of the peak current of the first return stroke of the cloud-to-ground lightning flashes across land/water boundaries over the contiguous United States. They showed a significant increase of the peak current exactly at the coastal boundaries especially during the convective wet summer season. They found that none of the inspected parameters (salinity of the ocean water, size of hydrometeors) could individually explain the land/ocean contrast. They speculated that an increased humidity might play an important role in the thundercloud microphysics and its influence on the thundercloud charging because of increased amounts of available charge in horizontally more extended clouds.

The occurrence of winter lightning can be also influenced by large scale climatic phenomena as the North Atlantic Oscillation (NAO) or the El Nino Southern Oscillation (ENSO). A positive NAO phase, which is characterized by an intensified Azores High and weakened Iceland Low, was found to lead to above-average precipitation and severe winter storms over British Isles and other parts of north-western and northern Europe including occurrence of extreme cyclones (Pinto et al., 2009). Lightning data from the Optical Transient Detector (OTD) from 1995 to 2000 for the North Atlantic Ocean and Western Europe were analyzed (de Pablo and Soriano, 2007) with respect to the NAO. The authors found a correlation between positive phase of NAO and increases of lightning rates at latitudes above 50°N. Williams et al. (2021) analyzed multi-station observations of Schumann resonances (SR) in order to investigate changes in the global lightning activity during two super El Niño events (1997 and 2014-2015). They found an increase of lightning activity in the transition from cold to warm phase during both events and confirmed their results deduced from the SR observations by independent analysis of the OTD and WWLLN data. They hypothesize that the increased occurrence of lightning can be caused by a thermodynamic disequilibrium between the surface and the middle troposphere during the transition.

The number of strokes per flash (flash multiplicity) is an interesting quantity, which is thought to reflect variations in climate and terrain (Schulz et al., 2005) through the changes in the amount of available charge in thunderclouds. The flash multiplicity is unfortunately very sensitive to both the detection efficiency of a given lightning location system and the algorithm used for grouping the strokes into a multi-stroke flash. Parameters, which determine inclusion of individual strokes into a flash, are the maximum inter-stroke distance, the maximum inter-stroke time interval and/or the maximum total duration of a flash. Slightly

different combinations of parameters were used in different studies: the maximum inter-stroke distance was 10 km, while the maximum inter-stroke interval (ISI) was either 0.5 s (Rakov & Huffines, 2003; Schulz et al., 2005; Pédeboy, 2012) or 1s (Cummins et al., 1998). A study of cold season lightning flashes using National Lightning Detection Network (NLDN) data (Adhikari & Liu, 2019) shows that 55 % of flashes contain only one stroke, 20 % had a multiplicity of 2, and remaining 25 % of flashes were composed of more than 2 strokes ( cold season is defined by 2-m surface temperature lower than 0 °C, ISI =1 s). About 16% of flashes were positive, with a larger fraction of single stroke flashes. In another study conducted in Austria with the EUCLID data from 1992 to 2001 (Schulz et al., 2005; ISI = 0.5 s), a multiplicity of negative lightning flashes was ~ 2.5 in contrast with a multiplicity of ~ 1.2-1.3 of positive lightning flashes. The global multiplicity of flashes detected by WWLLN was investigated using a comparison of WWLLN data and observation of the Lightning Imaging Sensor (LIS, on the Tropical Rainfall Measuring Mission satellite) in 2012-2014 (Burgesser, 2017). Spatially adjacent pixels exceeding the background threshold during one 2-ms window on the LIS sensors are clustered in groups. A set of groups, which are separated in time by no more than 330 ms and in space by no more than 5.5 km, is defined as a flash. It was found that WWLLN can detect multiple strokes during one LIS flash, when 70% of the matched LIS flashes were coincident with a single WWLLN stroke. A mean time difference and distance of 114 ms and 10 km, was respectively found between consecutive strokes multiple-stroke WWLLN flashes. Based on this finding, the global multiplicity of flashes detected by WWLLN estimated to 1.5 (Burgesser, 2017). This value is not directly comparable with the multiplicity derived from measurements of commercial lightning detection networks as NLDN or EUCLID because of lower density of WWLLN sensors, lower frequency range, and different grouping algorithm.

It is obvious from previous studies that occurrence of winter lightning in Europe at latitudes above 50°N is rare, and that spatial and temporal properties of winter lightning might reflect changes in global climate phenomena as NAO and ENSO which disrupt normal weather patterns. Nevertheless, a detailed analysis of properties of eastern North Atlantic winter lightning is still missing. In the present paper we report results of our analysis of lightning detected by WWLLN in the eastern North Atlantic and Northern Europe region during the winter 2014/2015, when the amount of winter lightning strokes exceeded three times a long-term average (four times a long-term median) calculated over the last decade (winter seasons 2010/2011 – 2019/2020). During the winter 2014/2015, UK, Germany, Poland, and Scandinavia suffered from extremely strong storms, which caused power outages in large areas, damages of buildings, and collapses of traffic paralyzing the daily life. An infrared image of the storm Rachel that threatened the UK and Ireland on 14 January 2015 was collected by the polar-orbiting NOAA19 satellite, and is shown in Fig. 1 as an example of a severe winter storm occurring in the analysed season. The occurrence of strong lightning was manifested by formation of a particular type of dispersed radio signals – so called daytime tweek atmospherics, which were found to originate in the north European lightning strokes. These usual night time signals were untypically observed during the day. After propagating in the sub-ionospheric waveguide, they were recorded at a low-noise observing site in the South of France and reported (Santolík and Kolmašová, 2017) for the first time in Europe.

In the present study, we investigate the temporal and spatial distribution of lightning flashes with respect to their energies and multiplicity. We especially focus on superbolts with energies above 1 MJ. Their peak currents reach above 3 MA when using an empirical formula 2 in Hutchins at al. (2012b), which was, however, derived considering all lightning and not specifically superbolts. Such extraordinary high peak currents, if they are real, would be by one order of magnitude larger than the highest lightning protection level (200 kA) recommended for protection of wind turbine rotor blades against lightning (Brondsted and Nijssen, 2013). Interestingly, this huge electromagnetic energy seems to have difficulties to leave the atmosphere: the superbolts with energies at least 1000 times larger than the mean energy of all lightning strokes detected by WWLLN were recently found by Ripoll et al., (2021) to transmit only 10–1000 times more powerful electromagnetic waves into the space in comparison with typical strokes. This discrepancy—, which has been unnoticed and unexplained— implies that remote sensing of superbolts from space might be useless and we have to rely on ground based observations.

In section 2, we describe our dataset. In section 3 we analyse properties of the whole dataset, including the characteristics of superbolts. In section 4 we show results on the stroke multiplicity, and in section 5 we discuss and summarize our findings.

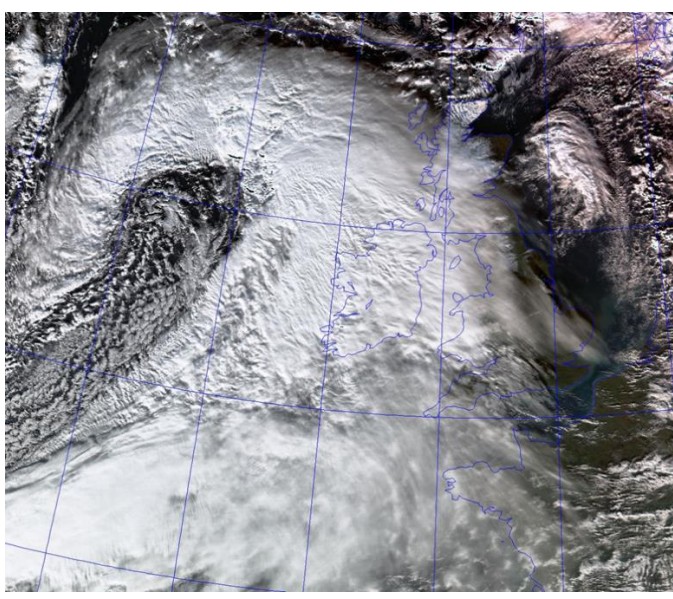

**Figure 1:** Infrared image collected by the polar-orbiting NOAA19 satellite shows the storm Rachel that threatened the UK and Ireland on 14 January 2015.

## 2 Dataset

WWLLN is an evolving global network of lightning location sensors operating in the VLF band at frequencies from 3 to 30 kHz with about seventy sensors in 2013 (Hutchins et al., 2013). WWLLN predominantly detects impulsive signals generated

by lightning return strokes called sferics, which can propagate thousands of kilometres in the Earth-ionosphere waveguide. WWLLN provides the time of occurrence and location of detected strokes. The location algorithm is based on the time of group arrival technique, which requires information from at least five WWLLN sensors. For the majority of strokes, WWLLN also delivers data about their energy with the uncertainty estimates. The information about the number of the WWLLN stations entering the algorithms for localization and energy estimation of individual strokes is also available. To determine the energy radiated by individual strokes, the root mean square (RMS) electric field of the triggered waveforms recorded at individual stations is used. The RMS electric field is calculated over a window of 1.33 ms in the 6–18 kHz band (Hutchins et al., 2012a). The U.S. Navy Long Wave Propagation Capability code (Ferguson, 1998) is used to model the propagation of the electromagnetic signal emitted by lightning strokes in the VLF band and to calculate the stroke energy needed to produce the measured RMS electric field values at individual WWLLN sensors. The energy radiated by individual strokes can be converted into the peak current using an empirical formula (equation 2 in Hutchins at al., 2012b), which was obtained by comparison of WWLLN detections and data from the New Zealand Lightning Location Network NZLDN. The polarity of the strokes cannot be derived from the WWLLN waveform measurements because the ground wave, which carries the information about the direction the current flowing in the lightning channel, is usually attenuated after travelling more than one thousand km from its source lightning to the receiver. WWLLN also provides a relative detection efficiency for each hour and each 1° x 1° bin (http://wwlln.net/deMaps). The detection efficiency changes in time not only because of the network upgrades, sensitivity of the sensors or data processing methods, but also during the day/night hours due to differences in the VLF wave propagation (Hutchins et al., 2012a). Kaplan and Lau (2021) analysed the raw WWLLN dataset from 2010-2020 and compared it with the data corrected by the relative detection efficiency maps. They found, that the annual stroke sum increased by about 11% from 2010–2013 by applying the detection efficiency coefficients, and that from 2014 the corrected and uncorrected datasets started to converge.

We used the WWLLN data (AE files) from 2010 to 2020 and determined the number of detections in ten winter seasons (October –March) in the area of our interest limited from the south by 50°N, and from the west by 20°W. The eastern boundary of 60°E was chosen to cover the northern part of the European continent up to the Ural Mountains. After applying the correction of raw lightning counts for each hour and each 1° x 1° bin by the detection efficiency of WWLLN, the stroke counts increased by 24 % in winter 2011/2012, by 9 % in 2010/2011, by 4% in 2019/2020, by 1.5 % in 2012/2013 and less than 0.5 % in other winter seasons. The resulting corrected annual amount of detections varied from 9,879 to 53,245 with an exception of the winter season 2014/2015 when WWLLN detected 103,323 lightning strokes. The average and median values of the yearly number of North-European winter strokes calculated over the last decade are respectively 36 and 25 thousand strokes. We selected the stormiest winter 2014/2015, and analyzed in detail spatial and temporal distribution of lightning strokes, their energies and multiplicity, focusing on extremely dangerous superbolts with megajoule energies.

After removing erroneous double detections from the originally distributed raw WWLLN data (102,866 strokes) (i.e., the strokes reported by mistake only tens of microseconds apart, personal communication R. Holzworth), our dataset consists of 92,132 localized individual lightning detections which occurred nearly every day between 1 October 2014 and 31 March 2015. We have identified only 14 days (out of 182) without any lightning activity in the whole area. If we use the same methodology as Taszarek et al. (2019) and calculate the number of thunderstorm days in 0.5° x 0.5° bins, we obtain bins with unusually high

(taking into account the winter season) numbers of thunderstorm days (TD) in the six individual investigated months (October: 7 TD, November: 9 TD, December: 13 TD, January: 11 TD, February: 5 TD, March: 6 TD). To create a subset of strokes with reliable energy estimates we excluded all cases with relative experimental energy uncertainties greater than 70%, and with energy estimates based on less than 3 stations. Applying these criteria for the reliability of energy estimates (Roger et al., 2017) we reduced the dataset by 17 %. The stroke energies ranged across five orders of magnitude from tens of J up to units of MJ,

with an asymmetric heavy-tail probability distribution. The strokes occurring in colder months of December, January and February (DJF) were ten times stronger (mean energy of 0.2 MJ with a standard deviation of 0.5 MJ and median energy of 6 kJ) than strokes hitting the North Atlantic in October, November and March (ONM) (mean energy of 0.02 MJ with a standard deviation of 0.1 MJ and median energy of 600 J). Fig. 2 illustrates the differences in energy distributions of strokes detected during colder and warmer winter months.

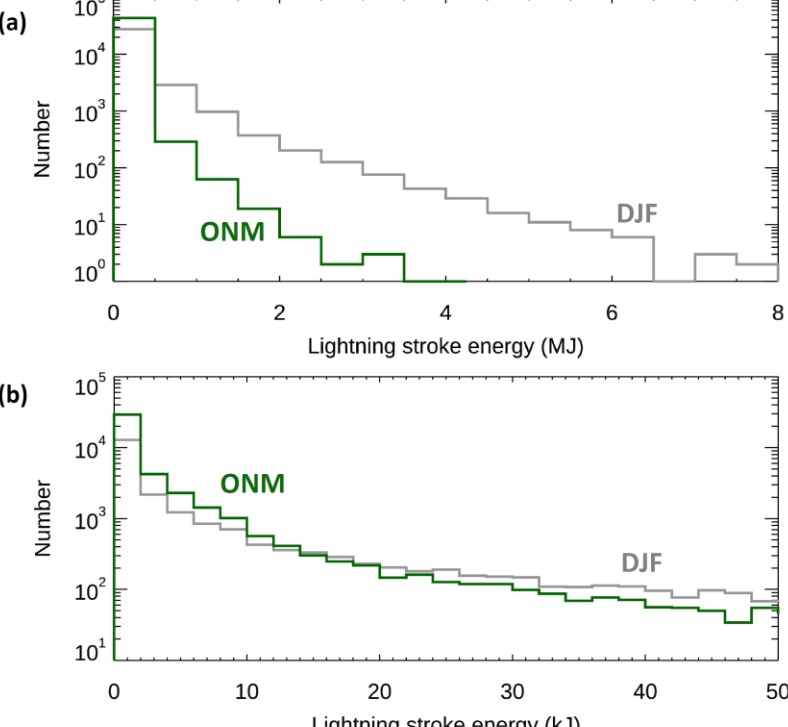

**Figure 2:** a) Histogram of energies of all lightning strokes detected by WWLLN during winter 2014-2015 in the North Atlantic region, bin size is 500 kJ, plotted separately for DJF (grey) and ONM (dark green). b) A detailed view on the weakest part of the stroke energy distribution with a bin size of 2 kJ

## 3 Spatial and temporal distribution of lightning strokes

The map on Fig. 3a shows the distribution of all detected lightning strokes plotted in 0.5° x 0.5° bins. They occurred predominantly above the ocean but with a higher concentration close to the western coastal areas, which were hit by up to 430 strokes per bin during the analysed period of 6 months, i.e., one stroke per 3.4 km$^2$ for the highest lightning density spot on the coast of Norway (62° N) and one stroke per 4.1 km$^2$ for the most intense spot on the coast of Denmark (55° N). The majority of lightning activity occurred above the sea: a rough estimate based on the NOAA Global Land One-kilometer Base Elevation data (doi:10.7289/V52R3PMS) gives only 21% lighting strokes over the land. This fraction is still likely overestimated because this method tends to randomly equalize the land/sea occurrences close to the highly rugged coastlines of Northern Europe, with the WWLLN spatial uncertainty of at least several km (Rodger et al., 2005).

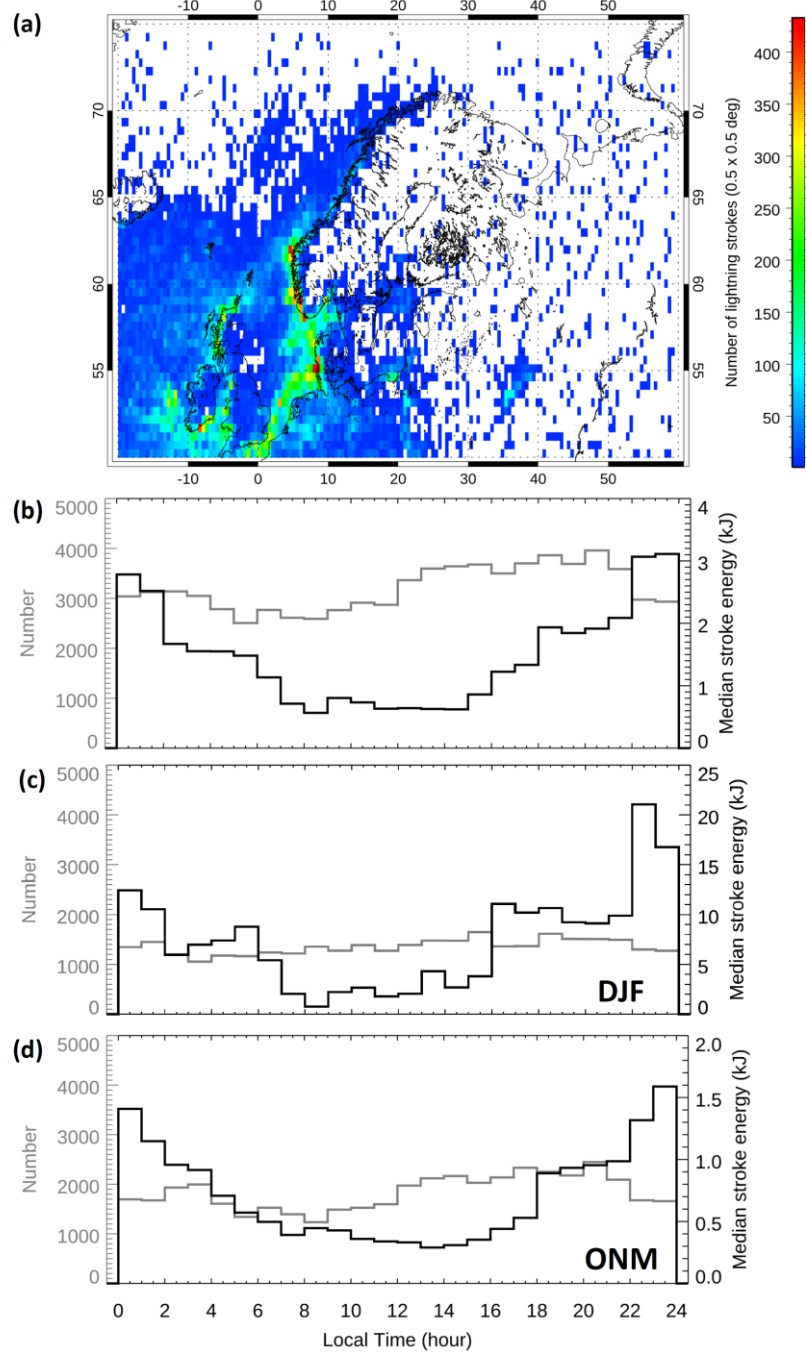

**Figure 3**: a) Spatial distribution of all detected lightning strokes in the 0.5° x 0.5° bins. b) Temporal distribution of all lightning strokes (grey line) and their hourly median energies (black line) as a function of the local time. The same as in (b), but for DJF (c) and ONM (d).

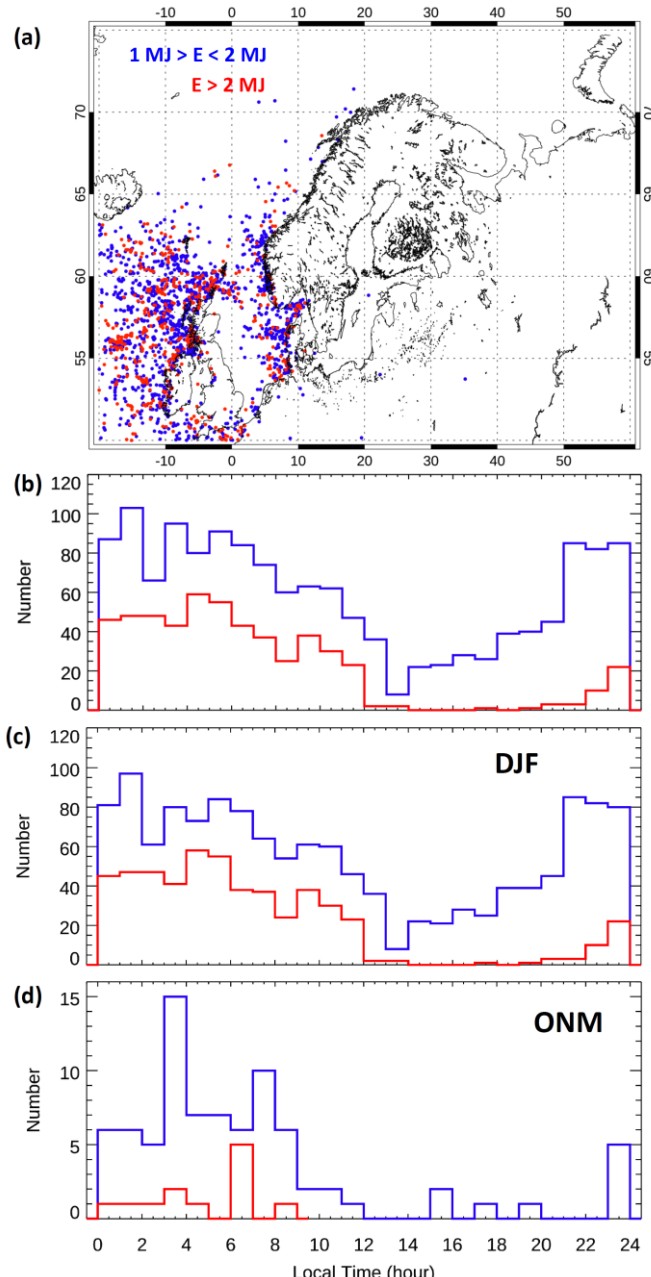

**Figure 4**: a) Spatial distribution of superbolts (strokes with energies between 1and 2 MJ and above 2 MJ are respectively represented by blue and red color). b) Temporal distribution of superbolts as a function of the local time. The same as in (b), but for DJF (c) and ONM (d).

The temporal distribution of lightning strokes with reliable energy estimates plotted as a function of the local time is represented in Fig. 3b by a grey line. The lightning discharges occurred nearly uniformly during the day and night and their

distribution did not exhibit a typical afternoon peak. Nevertheless, if we calculated the median energy in 1-hour local time bins (shown by a black line in Fig. 3b), a surprising peak arose around the local midnight. The strokes detected during the night had median energy values of about 3 kJ, three times higher than during the day. The energy median was calculated over 6 months. This effect is possibly even underestimated as the signals generated by daytime lightning are more attenuated when propagating in the Earth-ionosphere waveguide and we can thus expect a lower number of reliably detected weak strokes during the day which would shift the median daytime energies to higher values. The same histograms for DJF and ONM respectively shown in figures 3c and 3d illustrate thirteen times stronger strokes in colder months with the largest median of 21 kJ and 1.6 kJ before local midnight for DJF and ONM, respectively. The distribution in ONM is flatter than in DJF. Around the local noon, the difference the median energy values in DJF and ONM is not so prominent (2 kJ for DJF and 0.5 kJ for ONM).

Now we limit our dataset to extraordinary strong lightning and selected only superbolts – lightning strokes with energies above 1 MJ. Superbolts represented only 2.6 % of detected strokes with reliable energy estimates. Similarly, as in (Holzworth et al., 2019) we analyzed separately superbolts with energies between 1 and 2 MJ and superbolts with energies above 2 MJ, which are respectively represented in the map in Fig. 4a by blue and red dots. The superbolts appeared exclusively above the seawater with higher occurrence rates close to the western coastline of British Islands, Norway and Denmark. A few superbolts were detected even at high latitudes above 65°N. Temporal distributions of superbolts in Fig. 4b clearly show that superbolts only rarely struck in the afternoon and that the most energetic strokes with energies above 2 MJ preferred to appear in the night and morning hours. The majority of superbolts occurred during the three coldest months in the middle of the winter season (figures 4c and 4d).

## 4 Flash multiplicity

To investigate the multiplicity of flashes detected by WWLLN, we analyzed our whole dataset (including strokes with unreliable energy estimate) to find multi-stroke flashes consisting of strokes with striking points closer than 10 km, the inter-stroke intervals below 500 ms, and occurring within 1 s. These criteria were selected as the strictest ones among the criteria applied by different lightning location networks (Rakov & Huffines, 2003; Schulz et al., 2005; Pédeboy, 2012; Cummins et al., 1998). This grouping procedure resulted in 83 % of single-stroke flashes and 17 % of multi-stroke flashes. The number of strokes in individual multi-stroke flashes varied from two to twelve. The multiplicity from our dataset is illustrated in Fig. 5a by grey columns. In the reduced dataset with reliable energies, the energy ratio of the second stroke and the first stroke E2/E1 in all multiple flashes varied over eight orders of magnitude (Fig. 5b, solid line) with a median value of 0.16. The energy ratio of the third stroke and the first stroke E3/E1 shows similar properties as E2/E1, just for a smaller number of cases, with the median value reaching 0.11.

When we limited our dataset only to energies above 1 MJ (yellow columns in Fig. 5a), we found a similar percentage of subsequent strokes but only very exceptionally with multiplicities larger than 3. We also found that the superbolts struck just once: in the rare cases when subsequent strokes occurred, they never reached the superbolt energies above 1 MJ. Median energy ratios E2/E1 and E3/E1 in multiple flashes with superbolts are therefore extremely low, respectively reaching only $4.10^{-4}$ and $3.10^{-4}$. This means that they are by nearly three orders of magnitude weaker than subsequent strokes collected from the entire data set.

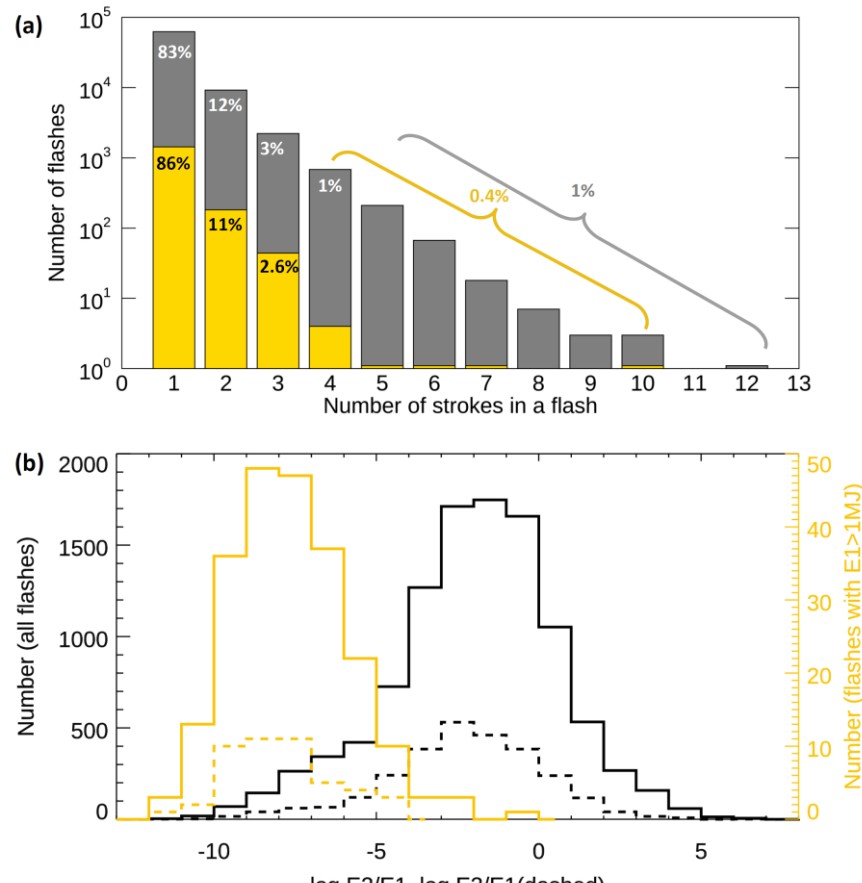

**Figure 5**: a) Multiplicity determined by a grouping algorithm applied on the whole dataset (grey), multiplicity of superbolts with the energy of the first stroke in a flash exceeding 1 MJ (yellow). b) Energy ratios of the second and the first stroke (solid line) and the third and the first stroke (dashed line) within all multiple flashes (black) and within superbolt flashes (gold).

## 5 Discussion and summary

WWLLN detected more than ninety thousand lightning strokes from October 2014 to March 2015. This quantity is four times higher than the median number of detected strokes calculated for the same seasons over the last decade (winter 2010/2011 – winter 2019/2020). There were 5-13 thunderstorm days in individual 0.5° x 0.5° bins and months calculated according to the methodology of Taszarek et al. (2019). The amount of bins with at least two detected discharges (thunderstorm days) was unusually large in comparison with the average number of 1-2 thunderstorm days in the same bin size reported by Taszarek et al. (2019) for the same region and the same season for years 2008-2017.

A median energy of 1.3 kJ for all detected strokes in our study is very close to a value of 1 kJ reported by Hutchins et al. (2012a) for the WWLLN global dataset from 2009-2012. However, the global mean value of 1 kJ related to the same dataset (Hutchins et al., 2012a) is by two order of magnitude smaller than in case of our winter dataset indicating a contribution of very energetic lightning strokes in the high energy tail of the distribution (especially in the colder months of December, January and February). The occurrence of winter superbolts in the eastern North Atlantic is expected (Holzworth et al., 2019). However, the comparison of our results with long-term lightning statistics demonstrates that the winter 2014/2015 was unusually rich on thunderstorm days with intense lightning activity.

Our analysis also shows that lightning predominantly occurred above the ocean and along the western coastal areas. This result is very different from the distribution of snow lightning (Adhikari & Liu, 2019). Nevertheless, it is consistent with the results of the global superbolt study (Holzworth et al., 2019), and we also show the same effect for weaker lightning. A rather surprising result is that the most energetic strokes appeared exclusively at night and in the morning hours and nearly 3 % of the detected lightning strokes were superbolts with an energy above 1 MJ.

When discussing the flash multiplicity, we have to take into account the sensitivity and relative detection efficiency of the particular lightning detection network. The relative detection efficiency of WWLLN is high in the investigated area because of dense network of sensors and a low attenuation of the signals propagating above the salty water. "We can assume that the relative efficiency did not change during the analyzed period in any of the 1° x 1° bins in the analyzed area, because the standard deviation of the daily relative efficiencies reached at maximum only 1.7% of their average value." As any other network, the sensitivity of WWLLN decreases for weak lightning. The comparison of WWLLN detections with the strokes detected by the American NLDN network showed (Abarca et al., 2010) that WWLLN detected at least 10% strokes with currents of ± 35 kA (707 J) and only about 1% of the weakest strokes with peak currents between −3 and −5 kA (13-30 J). We assume that in Europe, the sensitivity to weaker lightning would be similar and very weak subsequent strokes might be not detected by the network, especially when occurring over the ocean. This effect would artificially increase the fraction of single stroke flashes in the WWLLN data set. Nevertheless, the obtained large fraction of single stroke flashes (83 % of all events)

is significantly higher than 70 % of single stroke flashes reported for WWLLN dataset globally in tropics and subtropics. We moreover show that the superbolts with energies above one mega joule were also more frequently single stroke flashes (86%) and that their subsequent strokes never reached mega joule energies. This effect may occur due to the fact that the total amount of the charge available in the thundercloud for the whole flash was mostly neutralized during the first stroke and other energetic strokes could not be produced shortly after the first one.

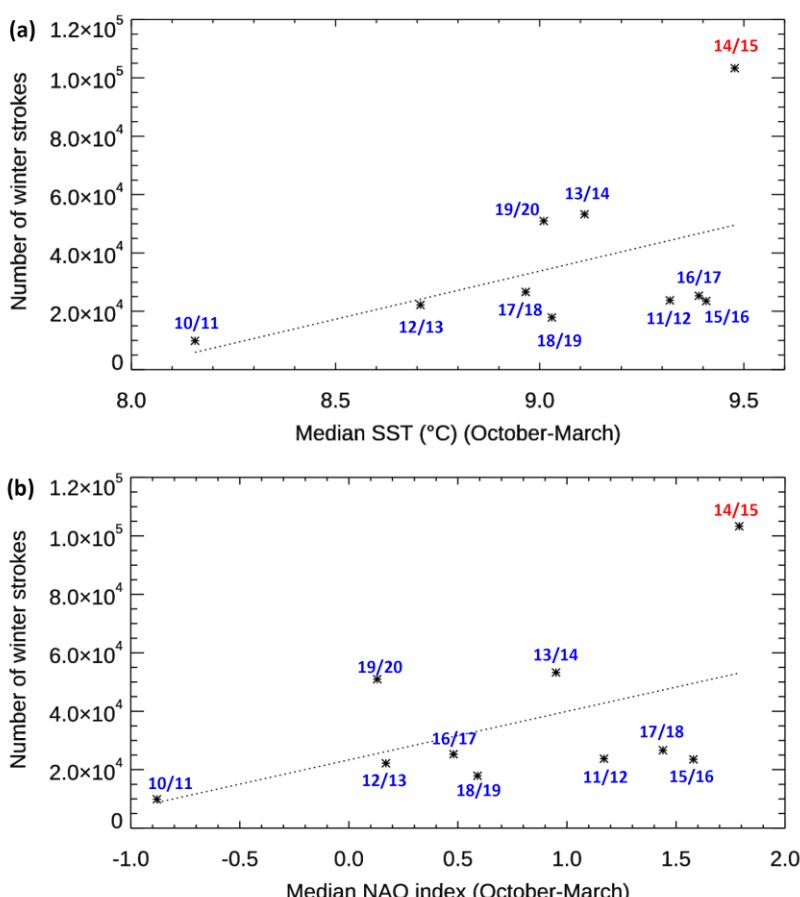

**Figure 6:** Number of lightning strokes detected by WWLLN in individual winter periods (October-March) and corrected by the relative detection efficiency maps as a function of a) the median winter SST in the eastern North Atlantic and b) the median NAO index calculated over the same period. The dashed lines show the linear trends.

All the above-mentioned findings indicate that unusually favourable conditions for a formation and electrification of thunderclouds might have arisen in winter 2014/2015. As a mixture of water and ice hydrometeors and their charging by
collisions are needed for a cloud electrification (Rakov and Uman, 2003) we speculate that there was an unusual amount of charged hydrometeors in winter thunderclouds. Such increased supply of available cloud charge might be due to: a) larger extent of thunderclouds, b) changes in cloud microphysics, c) higher updraft speeds. As the first step, we checked  surface sea temperatures (SST) in the eastern North Atlantic region as their increase would allow setting of the typical meteorological scenario for winter lightning (cold air overblowing a warmer seawater; Williams, 2018). Such SST anomalies were found to
occur during El Nino events (Williams et al., 2021) or during the positive phase of the North Atlantic Oscillation (Qu et al., 2012). We analyse the monthly SST data provided by the IRI/LDEO collection of climate data in a 1° grid (NOAA NCEP EMC CMB GLOBAL Reyn_SmithOIv2; Reynolds et al., 2002). We calculated the mean monthly sea surface temperatures for winter seasons 2010-2020 in an area limited by coordinates 20ºW, 10ºE, 50ºN, 65ºN, where we detected most discharges in winter 2014/2015 (Figs. 3a and 4a). Then we plot total number of lightning strokes as a function of median SSTs calculated
for individual winter seasons (Fig. 6a). The median winter temperatures varied from 8 to 9.5ºC in the selected area. We can see an increase of lightning activity with increasing SSTs, but the spread of values is very large. During the stormy winter season 2014/2015, when the El Nino event started, the sea surface was unusually warm reaching a temperature of 9.48ºC, the highest winter value within the last decade. Nevertheless, the temperatures during the winter seasons 2015/2016 (El Nino peak) and 2016/2017 (El Nino warm to cold transition phase) were only very slightly lower with a substantially weaker lightning
activity. We have also checked the variations of NAO monthly indexes during last two decades provided by the Climate Prediction Center of the Weather National Service NOAA. The monthly NAO index describes the strength of NAO.  Its calculation is based on the difference between normalized mean sea-level pressure strengths of the Azores High and the Icelandic Low.The median values for last twenty winter seasons (October – March) vary from -1.2 to 1.4. The winter season 2014/2015 exhibited the highest median positive value of the NAO within two last decades, and the NAO indexes reached
even 1.9 in December 2014 and 1.8 in January 2015. If we plot the numbers of lightning strokes detected in last ten winter seasons as a function of the median NAO indexes calculated for individual winter seasons from monthly NAO indexes (Fig.6b), we can see an increase of lightning activity with an increasing NAO index despite the large variance of values. There were more strokes detected in winter 2014/2015 than it might be expected from both trends in Figs. 6a and 6b.

In summary, numerous energetic lightning hit the British Islands and Northern Europe in winter 2014/2015. We found that a documented unusually warm sea surface in the eastern North Atlantic and a large positive NAO index do not themselves explain the anomalously intense winter lightning activity. It is evident that an additional driver was acting in the cloud electrification.  A good candidate is an increased updraft velocity, as the lightning rate was found to be characteristically proportional to the sixth power of the updraft speed (Baker et al., 1999). A strong updraft originating in a thermodynamic
disequilibrium due the contrast in temperatures between land and ocean was suggested by Williams et al. (2021) to accompany the transition from the cold to warm phase of El Nino especially at the ocean/sea boundary. Above-mentioned factors might

have changed the typical winter microphysical thundercloud composition, charging processes, and/or electrical conductivity, and could result in a more efficient cloud electrification (Chronis et al, 2016). The evidence in favour of this hypothesis is the observed spatial distribution of lightning. The strokes were concentrated above the ocean and along the western coastal areas with a quite sharp change in the lightning density at the ocean/land boundary. The strokes were very energetic; the mean stroke energy calculated over the whole period of six months was by two orders of magnitude larger than the global WWLLN mean energy of 1 kJ. Three per cent of the detected strokes were the superbolts with an energy exceeding 1 MJ. The most energetic strokes occurred during night and morning hours when there could be a larger vertical temperature gradient in the atmosphere. We estimated for the first time the multiplicity of winter strokes in the WWLLN data using the grouping criteria similar to other lightning location networks. We have found that more flashes than in other seasons and regions were single-stroke flashes and that subsequent strokes in superbolts never reached mega joule energies. Global climate events occurring during analysed period were probably responsible for the increase of the sea surface temperature, significantly stronger updraft velocities, and for the consequent intense energetic lightning activity: the significant positive phase of NAO and the transition from cold to warm phase of ENSO. Our study therefore indicates that local distribution of lightning can reflect the disruptions in normal weather patterns originating in global climate phenomena. Based on the distribution, strength and intensity of lightning —and especially superbolts— presented in our study, our findings might have an impact on the applications of lightning protection measures because of the rapidly increasing number of wind turbines and existing oil platforms in the North Sea.

**Data availability**

WWLLN archival data are copyrighted by the University of Washington and are available to the public at nominal cost (wwlln.net). The monthly mean NAO indexes are available at https://www.cpc.ncep.noaa.gov/products/precip/CWlink/pna/norm.nao.monthly.b5001.current.ascii.

**Author contribution**

OS and IK designed the study, interpreted the results and wrote the paper. IK and KR performed the data analysis.

**Competing interests**

The authors declare that they have no competing interests.

## Acknowledgements

The work of IK, OS and KR was supported by the GACR grant 20-09671S and by European Regional Development Fund-Project CRREAT (CZ.02.1.01/0.0/0.0/15_003/0000481). We are grateful to the NEODAAS/University of Dundee for providing the NOAA19 infrared image.

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
