# Peer review of "Lightning activity in Northern Europe during a stormy winter: disruptions of weather patterns originating in global climate phenomena"

_Atmospheric Chemistry and Physics, 2021_

## Referee Comment (RC1)

The manuscript by Ivana Kolmasova and colleagues analyses lightning occurrences over the northern Atlantic and the North Sea during the winter season 2014/2015. It is found that intense lightning discharges, named superbolts, are predominantly found at nighttime in the morning hours as single stroke flashes. The manuscript is extremely well written, easy to follow, logically constructed and the figures support the text. The content of the paper is enjoyable to read. Particularly interesting is the explanation of the authors how the North Atlantic Oscillation might have contributed to the production of more severe electrified storms when compared to other meteorological processes that produce abundant lightning. Although not specifically mentioned in the manuscript, the findings might have a significant impact by informing lightning protection measures for the rapidly increasing number of wind turbines in the North Sea and existing oil platforms. As a result, I have only a few very minor comments to improve the quality of the manuscript which the authors might want to consider prior to publication.

(1) l119 - Explain what is meant by 'huge'. Relatively large spatial scale or long duration, or both, or number of customers affected?
(2) l129 - The excellent and exhaustive literature review could be supplemented by a recent publication from Ripoll et al., in Nature Communications, 2021, for comparison.
(3) l142 - Clarify whether the ~70 stations existed now, or already in 2014/2015.
(4) l163 - Add the southern boundary and explain why an extension to 60 deg East is of interest, which seems far from the northern Atlantic and the North Sea. Use 9,066 and 53,182.
(5) l167 - Clarify what is dangerous about superbolts, when compared to lightning protection guidelines, if that is what is meant.
(6) l168 - 10,742
(7) l252 - Replace the acronym 'TD' by thunderstorm days, to enhance readability.
(8) l272 - change
(9) l301 - Explain in one sentence what the NAO index measures for the unfamiliar reader.

---

## Referee Comment (RC3)

Review of the manuscript "Lightning activity in Northern Europe during a stormy winter: disruptions of weather patterns originating in global climate phenomena" by Ivana Kolmašová, Ondřej Santolík, and Kateřina Rosická (acp-2021-827).

**General comments.**

The manuscript analyses the lightning activity, detected by WWLLN, during the 2014/2015 colder season over the northern Atlantic. The period of time analyzed was chosen given that this season presented an unusually high lightning activity with more than 5 thunderstorm days per month.

The analysis shows that lightning occurs predominantly above the ocean and along the western coastal areas with a nearly uniform (local) time occurrence and a flash multiplicity between 1 and 12 but with an 86% of the strokes detected by WWLLN as a single-stroke flashes.

On the other hand, the analysis of the superbolts (defined as lightning strokes with energies above 1 MJ) shows that this type of lightning strokes occurs above seawater on the western coastline of British Islands, Norway and Denmark, with the majority of the superbolts occurring during the three coldest months in the middle of the winter season. The (local) time distribution of the superbolts shows a preference to occur at night and morning hours. The analysis of the superbolts multiplicity showed that 86% of the superbolts are single-stroke flash and superbolts only present multiplicity up to 3. The analysis of the energy of successive strokes forming a flash shows that, after a very energetic strokes (superbolt), the subsequent strokes have a three orders weaker energy magnitude.

Finally, the unusually high lightning activity detected during the 2014/2015 winter is discuss in term of the North Atlantic Oscillation and the variation of the NAO monthly index.

The manuscript present a novelty analysis of the lightning activity produced by winter thunderstorm and on an area were the lightning activity is rare. The analysis provides a new insight on the distribution of high energy strokes which same to be related with the SST anomalies reported on the northern Atlantic.

The manuscript is well written and the presentation is well structured and clear. The figures are adequate and support the analysis and results presented on the study.

However, the authors should discuss some issues before the manuscript is ready for publication.

**Major comments.**

1. Lines 162-165. The authors indicated that they used the WWLLN data between 2010 and 2020 on the area under study. According to the authors, the WWLLN annual amount of detections varied through this period with a peak on the lightning strokes detected during the winter season 2014/2015. However, the WWLLN detection efficiency has change through the years caused by the buildout of the network. As showed by Kaplan and Lau (2021), the amount of strokes detected by WWLLN had increased since 2005 until 2014 when the network seems to reach a stable detection amount of strokes.

Had been the amount of lightning strokes, reported by the authors, corrected by the network efficiency?

If not, perhaps the peak observed on the winter season 2014/2015 is not so high as the study suggest given the detection efficiency time variation.

On lines 165-167, the authors also reported that "The average and median values of the yearly number of North-European winter strokes calculated over the last decade are respectively 35 and 24 thousand strokes."

Has been these average and median values calculated with the raw data reported by WWLLN? If the average and median values were calculated with the raw data reported by WWLLN, the results with have a bias given the low detection amount of lightning strokes during the first part of the decade. The detection efficiency variation of the WWLLN needs to be discuss on the manuscript. Kaplan, J. O., & Lau, K. H. K. (2021). The WGLC global gridded lightning climatology and timeseries. Earth System Science Data Discussions, 1-25.

2. Lines 177-182. The authors reported that the strokes occurred during DJF were ten time stronger than the strokes occurred during ONM. This conclusion is based on the values of the mean and median of the energy distribution for these two period of time. However, the strokes energy presents a nearly log-normal distribution and the mean and median are not the best way to compare these type of energy distributions. Please show the energy distribution of the lightning strokes for both period of time in the same graph to compare and discuss the differences.

3. Lines 228-229. Please discuss how was decide the grouping criteria chosen to define a multi-stroke flash. Were these criteria based on the literature or sensitivity tests were made to define its?

4. Lines 272-273. The authors asseverated that the relative efficiency of the network did not change during the analyzed period. Is this an assumption or is based on an analysis of the efficiency reported by the network? Please clarify.

5. Lines 299-301. The authors refer to a global SST anomaly map available at NASA web page. The link shows a video for a long timeline and it is not easy to see the SST anomaly for the norther Atlantic for the 2014/2015 winter season. Please consider to present a new figure showing the SST anomaly for the period and area of interest.

6. Lines 305 and Figure 6. It is reported the winter season numbers of lighting strokes detected by WWLLN as a function of the NOA index. However, as I mention before (see comment 1), the WWLLN detection efficiency varies through the years. Therefore, the high amount of lightning activity observed in the 2014/2015 winter could have a bias given the low detection efficiency during the first part of the decade.

**Minor corrections:**

1. Lines 174-175. "...months (7, 9, 13, 11, 5, and 6, respectively). To create...". Please clarifies the meaning of the number between the parenthesis.

2. Line 194. Please indicate how was obtained the values of 1 strokes per 3.3 km2 reported.

3. Lines 194-195. "...There was nearly no lightning activity detected above the continent." Please report the percentage of the amount of the strokes detected over the ocean and the continent.

4. Figure 3 b, c and d and Figure 4 b, c and d. Please considerer to use the same vertical axis in the graphics.

5. Figure 5 a. The sum of the percentage reported on the first bins is 100%. What percentage represent the other bins reported (multiplicity higher than 4)?

---

## Author Comment (AC1)

*We thank the reviewer for her/his careful reading of our manuscript and for her/his additional comments. We highly appreciate the time she/he invested into helping us to improve the paper.*

*Responses to Reviewer #1 (in blue) with Reviewers´ comments in black.*
* * *
The manuscript by Ivana Kolmasova and colleagues analyses lightning occurrences over the northern Atlantic and the North Sea during the winter season 2014/2015. It is found that intense lightning discharges, named superbolts, are predominantly found at nighttime in the morning hours as single stroke flashes. The manuscript is extremely well written, easy to follow, logically constructed and the figures support the text. The content of the paper is enjoyable to read. Particularly interesting is the explanation of the authors how the North Atlantic Oscillation might have contributed to the production of more severe electrified storms when compared to other meteorological processes that produce abundant lightning. Although not specifically mentioned in the manuscript, the findings might have a significant impact by informing lightning protection measures for the rapidly increasing number of wind turbines in the North Sea and existing oil platforms. As a result, I have only a few very minor comments to improve the quality of the manuscript which the authors might want to consider prior to publication.

Thanks to the recommendation of the reviewer, we added following sentence related to lightning protection at the end of the summary section. "*Based on the distribution, strength and intensity of lightning —and especially superbolts— presented in our study, our findings might have an impact on the applications of lightning protection measures because of the rapidly increasing number of wind turbines and existing oil platforms in the North Sea.* "

(1)l119 - Explain what is meant by 'huge'. Relatively large spatial scale or long duration, or both, or number of customers affected?

By huge power outages, we meant both large areas and number of customers affected. We modified the text accordingly and now it reads as follows on line 120:

„*During the winter 2014/2015, UK, Germany, Poland, and Scandinavia suffered from extremely strong storms, which caused power outages in large areas, damages of buildings, and collapses of traffic paralyzing the daily life.*"

A few links illustrating the exceptionality of the weather conditions are below:

https://www.welt.de/vermischtes/weltgeschehen/article136239231/Das-verrueckteste-Wetter-seit-100-Jahren.html

https://www.dailyrecord.co.uk/news/scottish-news/thousands-homes-north-scotland-remain-4961335

https://www.express.co.uk/news/nature/551859/UK-weather-latest-thundersnow-Britain-Storm-Rachel

https://www.theguardian.com/uk-news/2015/jan/11/thousands-homes-without-power-scotland-uk-weather-warning

(2)l129 - The excellent and exhaustive literature review could be supplemented by a recent publication from Ripoll et al., in Nature Communications, 2021, for comparison.

We added the recommended reference and corresponding text on lines 135-138:

"*Interestingly, this huge electromagnetic energy seems to have difficulties to leave the atmosphere: the superbolts with energies at least 1000 times larger than the mean energy of all lightning strokes detected by WWLLN were recently found by Ripoll et al., (2021) to transmit only 10–1000 times more powerful electromagnetic waves into the space in comparison with typical strokes. This discrepancy—, which has been unnoticed and unexplained— implies that remote sensing of superbolts from space might be useless and we have to rely on ground based observations.*"

(3)l142 - Clarify whether the ~70 stations existed now, or already in 2014/2015.
There have been about 70 WWLLN sensors operational already since 2013 as reported in Hutchins et al. (2013). We added the information to the text on line 152 and added the relevant reference.

(4)l163 - Add the southern boundary and explain why an extension to 60 deg East is of interest, which seems far from the northern Atlantic and the North Sea. Use 9,066 and 53,182.
The southern boundary is 50°N, it is already mentioned in the manuscript. The norther boundary was not mentioned, as it is 90°N. The eastern boundary was set to 60°E to cover the northern part of European continent up to the border between Europe and Asia. We modified the text accordingly on lines 172-173: " …. the area of our interest limited from the south by 50°N, and from the west by 20°W. The eastern boundary of 60°E was chosen to cover the northern part of the European continent up to the Ural Mountains."
The format of numbers was changed.

(5)l167 - Clarify what is dangerous about superbolts, when compared to lightning protection guidelines, if that is what is meant.
To illustrate hazards related to superbolts we added following text on lines 131-135: "*Their peak currents reach above 3 MA when using an empirical formula 2 in Hutchins at al. (2012b), which was, however, derived considering all lightning and not specifically superbolts. Such extraordinary high peak currents, if they are real, would be by one order of magnitude larger than the highest lightning protection level (200 kA) recommended for protection of wind turbine rotor blades against lightning (Brondsted and Nijssen, 2013). "*

 (6)l168 - 10,742
Done on line 180.

(7)l252 - Replace the acronym 'TD' by thunderstorm days, to enhance readability.
Done on line 262.

(8)l272 - change
Done on line 282.

(9)l301 - Explain in one sentence what the NAO index measures for the unfamiliar reader.
We added an explanation on lines 313-314: „*The monthly NAO index describes the strength of NAO. Its calculation is based on the difference between normalized mean sea-level pressure strengths of the Azores High and the Icelandic Low.*"

---

## Author Comment (AC2)

We thank Deborah Morgenstern for hers careful reading of our manuscript and for her comments. We highly appreciate the time she invested into helping us to improve the paper.

**Responses (in blue) with Deborah's comments in black.**

Thanks for your interesting analysis.

Regarding your conclusion about the higher sea surface temperatures and increased water vapor (Abstract line 24 and lines 294-301):

Could you please describe more precisely what you have analyzed and how the SST changed in your observed region? You could replace the link to this clip with global SST by some numbers for your observed region or even produce a plot similar to Fig.6.

We followed the recommendation, and analyzed thoroughly the changes in the sea surface temperatures and added a panel in Fig. 6. Now the number of lightning strokes detected by WWLLN in individual winter periods (October-March) is plotted as a function of a) the median winter sea surface temperatures in the eastern North Atlantic and b) the median NAO index calculated over the same period. The method is now described on lines 312-320 and reads:

"We analyze the monthly SST data provided by the IRI/LDEO collection of climate data in a 1° grid (NOAA NCEP EMC CMB GLOBAL Reyn\_SmithOlv2; Reynolds et al., 2002). We calculated the mean monthly sea surface temperatures for winter seasons 2010-2020 in an area limited by coordinates 20°W, 10°E, 50°N, 65°N, where we detected most discharges in winter 2014/2015 (Figs. 3a and 4a). Then we plot total number of lightning strokes as a function of median SSTs calculated for individual winter seasons (Fig. 6a). The median winter temperatures varied from 8 to 9.5°C in the selected area. We can see an increase of lightning activity with increasing SSTs, but the spread of values is very large. During the stormy winter season 2014/2015, when the El Nino event started, the sea surface was unusually warm reaching a temperature of 9.48°C, the highest winter value within the last decade. Nevertheless, the temperatures during the winter seasons 2015/2016 (El Nino peak) and 2016/2017 (El Nino warm to cold transition phase) were only very slightly lower with a substantially weaker lightning activity."

We added a paragraph in the discussion on lines 305-308, which now reads:

"As a mixture of water and ice hydrometeors and their charging by collisions are needed for a cloud electrification (Rakov and Uman, 2003) we speculate that there was an unusual amount of charged hydrometeors in winter thunderclouds. Such increased supply of available cloud charge might be due to: a) larger extent of thunderclouds, b) changes in cloud microphysics, c) higher updraft speeds. As the first step, we checked surface sea temperatures (SST) in the eastern North Atlantic region as their increase would allow setting of the typical meteorological scenario for winter lightning (cold air overblowing a warmer seawater; Williams, 2018). Such SST anomalies were found to occur during El Nino events (Williams et al., 2021) or during the positive phase of the North Atlantic Oscillation (Qu et al., 2012)."

Figure 6: Number of lightning strokes detected by WWLLN in individual winter periods (October-March) as a function of a) the median winter sea surface temperatures in the eastern North Atlantic and b) the median NAO index calculated over the same period. The dashed lines shows the linear trends.

Please support your speculation about enhanced available atmospheric water vapor by data or omit this statement from your manuscript.

We followed the hypothesis of Williams et al, 2021 that during the could—to-warm transition phase of El Nino there probably was a stronger updraft leading to very efficient electrification and anomalous lightning rates. We added a sentence into the introduction on lines 89-90 and a reference to Baker et al. (1999) who showed that the lightning rate is proportional to the sixth power of the updraft speed.

"Williams et al. (2021) hypothesize that the increased occurrence of lightning can be caused by a thermodynamic disequilibrium between the surface and the middle troposphere during the transition."

We also modified the abstract and the summary paragraph.

The abstract now ends by following sentences:

"All these lightning characteristics suppose an anomalously efficient winter thundercloud charging in the eastern North Atlantic, especially at the sea-land boundary. We found that the resulting unusual production of lightning could not be explained solely by an anomalously warm sea surface caused by a positive phase of the North Atlantic Oscillation and by a starting super El Nino event. Increased updraft strengths, which are believed to accompany the cold to warm transition phase of El Nino, might have acted as another charging driver. We speculate that a combination of both these large-scale climatic events might have been needed to produce observed enormous amount of winter lightning in winter 2014/2015."

The summary part now started on lines 330-336 as follows:

"In summary, numerous energetic lightning hit the British Islands and Northern Europe in winter 2014/2015. We found that a documented unusually warm sea surface in the eastern North Atlantic and a large positive NAO index do not themselves explain the anomalously intense winter lightning activity. It is evident that an additional driver was acting in the cloud electrification. A good candidate is an increased updraft velocity, as the lightning rate was found to be characteristically proportional to the sixth power of the updraft speed (Baker et al., 1999). A strong updraft originating in a thermodynamic disequilibrium due the contrast in temperatures between land and ocean was suggested by Williams et al. (2021) to accompany the transition from the cold to warm phase of El Nino especially at the ocean/sea boundary. Above-mentioned factors might have changed the typical winter microphysical thundercloud composition, charging processes, and/or electrical conductivity, and could result in a more efficient cloud electrification (Chronis et al, 2016)."

---

## Author Comment (AC4)

**We thank the reviewer for her/his careful reading of our manuscript and for her/his additional comments. We highly appreciate the time she/he invested into helping us to improve the paper.**

**Responses to Reviewer #3 (in blue) with Reviewers' comments in black.**

Review of the manuscript "Lightning activity in Northern Europe during a stormy winter: disruptions of weather patterns originating in global climate phenomena" by Ivana Kolmašová, Ondřej Santolík, and Kateřina Rosická (acp-2021-827). General comments.

The manuscript analyses the lightning activity, detected by WWLLN, during the 2014/2015 colder season over the northern Atlantic. The period of time analyzed was chosen given that this season presented an unusually high lightning activity with more than 5 thunderstorm days per month.

The analysis shows that lightning occurs predominantly above the ocean and along the western coastal areas with a nearly uniform (local) time occurrence and a flash multiplicity between 1 and 12 but with an 86% of the strokes detected by WWLLN as a single-stroke flashes.

On the other hand, the analysis of the superbolts (defined as lightning strokes with energies above 1 MJ) shows that this type of lightning strokes occurs above seawater on the western coastline of British Islands, Norway and Denmark, with the majority of the superbolts occurring during the three coldest months in the middle of the winter season. The (local) time distribution of the superbolts shows a preference to occur at night and morning hours. The analysis of the superbolts multiplicity showed that 86% of the superbolts are single-stroke flash and superbolts only present multiplicity up to 3. The analysis of the energy of successive strokes forming a flash shows that, after a very energetic strokes (superbolt), the subsequent strokes have a three orders weaker energy magnitude.

Finally, the unusually high lightning activity detected during the 2014/2015 winter is discuss in term of the North Atlantic Oscillation and the variation of the NAO monthly index. The manuscript present a novelty analysis of the lightning activity produced by winter thunderstorm and on an area were the lightning activity is rare. The analysis provides a new insight on the distribution of high energy strokes which same to be related with the SST anomalies reported on the northern Atlantic.

The manuscript is well written and the presentation is well structured and clear. The figures are adequate and support the analysis and results presented on the study. However, the authors should discuss some issues before the manuscript is ready for publication.

**Major comments.**

1. Lines 162-165. The authors indicated that they used the WWLLN data between 2010 and 2020 on the area under study. According to the authors, the WWLLN annual amount of detections varied through this period with a peak on the lightning strokes detected during the winter season 2014/2015. However, the WWLLN detection efficiency has change

through the years caused by the buildout of the network. As showed by Kaplan and Lau (2021), the amount of strokes detected by WWLLN had increased since 2005 until 2014 when the network seems to reach a stable detection amount of strokes. Had been the amount of lightning strokes, reported by the authors, corrected by the network efficiency? If not, perhaps the peak observed on the winter season 2014/2015 is not so high as the study suggest given the detection efficiency time variation.

The number of lightning strokes was not corrected for efficiency in the previous version of the manuscript. Thanks to the reviewer's comment we now systematically use this correction. Specifically, we corrected the raw counts of the lightning strokes in the investigated area and periods, directly using the detection efficiency maps provided by WWLLN. The obtained corrected counts did not decrease the 2014/2015 peak, which remained 4 times higher than the median count calculated over the last decade. The text with corrected numbers appears on lines 185-186: *" The resulting corrected annual amount of detections varied from 9,879 to 53,245 with an exception of the winter season 2014/2015 when WWLLN detected 103,323 lightning strokes."*

On lines 165-167, the authors also reported that "The average and median values of the yearly number of North-European winter strokes calculated over the last decade are respectively 35 and 24 thousand strokes." Has been these average and median values calculated with the raw data reported by WWLLN? If the average and median values were calculated with the raw data reported by WWLLN, the results with have a bias given the low detection amount of lightning strokes during the first part of the decade.

We corrected the average and median values of strokes using the data corrected by the detection efficiency maps. The new average value of the counts of winter lightning calculated over last decade increased by 826 strokes and the new median value increased by 145 strokes. After rounding to thousands, both numbers increased by 1 thousand and the text now reads on line 187 as follows: *"The average and median values of the yearly number of North-European winter strokes calculated over the last decade are respectively 36 and 25 thousand strokes."*

The detection efficiency variation of the WWLLN needs to be discuss on the manuscript.

The detection efficiency was applied on the raw data and resulting differences in stroke counts were added on lines 182-185: "After applying the correction of raw lightning counts for each hour and each 1° x 1° bin by the detection efficiency of WWLLN, the total *stroke counts increased by 24 % in winter 2011/2012, by 9 % in 2010/2011, by 4% in 2019/2020, by 1.5 % in 2012/2013 and less than 0.5 % in other winter seasons."*

Kaplan, J. O., & Lau, K. H. K. (2021). The WGLC global gridded lightning climatology and timeseries. Earth System Science Data Discussions, 1-25.

We added the paper in the list of reference and relevant text on lines 175-178: "Kaplan and Lau (2021) analysed the raw WWLLN dataset from 2010-2020 and compared it with the data corrected by the relative detection efficiency maps. They found, that the annual stroke sum

**increased by about 11% from 2010–2013 by applying the detection efficiency coefficients, and that from 2014 the corrected and uncorrected datasets started to converge."**

2. Lines 177-182. The authors reported that the strokes occurred during DJF were ten time stronger than the strokes occurred during ONM. This conclusion is based on the values of the mean and median of the energy distribution for these two period of time. However, the strokes energy presents a nearly log-normal distribution and the mean and median are not the best way to compare these type of energy distributions. Please show the energy distribution of the lightning strokes for both period of time in the same graph to compare and discuss the differences.

Instead of energy distributions of all winter strokes, we show DJF and ONM stroke energies separately in Fig. 2, to illustrate differences in the energy distribution of strokes occurring in colder and warmer winter months. We also changed the properties of the probability distribution for stroke energies from log-normal to heavy-tail one on line 201.

New Figure 2.

3. Lines 228-229. Please discuss how was decide the grouping criteria chosen to define a multi-stroke flash. Were these criteria based on the literature or sensitivity tests were made to define its?

We add a sentence on lines 263-265. *"These criteria used for grouping the strokes in multi-stroke flashes in our study were selected as the strictest ones among the criteria applied by different lightning location networks (Rakov & Huffines, 2003; Schulz et al., 2005; Pédeboy, 2012; Cummins et al., 1998).*

4. Lines 272-273. The authors asseverated that the relative efficiency of the network did not change during the analyzed period. Is this an assumption or is based on an analysis of the efficiency reported by the network? Please clarify.

The assumption was based on the analysis, the relevant text was added on lines 288 - 290 and reads as follows: "We can assume, that the relative efficiency did not change during the analyzed period in any of the  $1^{\circ} \times 1^{\circ}$  bins in the analyzed area, because the standard deviation of the daily relative efficiencies reached at maximum only 1.7 % of their average value."

5. Lines 299-301. The authors refer to a global SST anomaly map available at NASA web page. The link shows a video for a long timeline and it is not easy to see the SST anomaly for the norther Atlantic for the 2014/2015 winter season. Please consider to present a new figure showing the SST anomaly for the period and area of interest.

The figure and relevant changes in the text were already added in the previous version of the manuscript, which took into account the community comment CC1. We analyzed thoroughly the changes in the sea surface temperatures using the monthly SST data provided by the IRI/LDEO collection of climate data in a 1° grid (NOAA NCEP EMC CMB GLOBAL Reyn\_SmithOlv2; Reynolds et al., 2002). ). We calculated the mean monthly sea surface temperatures for winter seasons 2010-2020 in an area limited by coordinates 20°W, 10°E, 50°N, 65°N, where we detected most discharges in winter 2014/2015 (Figs. 3a and 4a). We added a panel in Fig. 6. Now the number of lightning strokes detected by WWLLN in individual winter periods (October-March) is plotted as a function of a) the median winter sea surface temperatures in the eastern North Atlantic and b) the median NAO index calculated over the same period.

New Figure 6.

6. Lines 305 and Figure 6. It is reported the winter season numbers of lighting strokes detected by WWLLN as a function of the NOA index. However, as I mention before (see comment 1), the WWLLN detection efficiency varies through the years. Therefore, the high amount of lightning activity observed in the 2014/2015 winter could have a bias given the low detection efficiency during the first part of the decade.

The figure 6 (shown above) was redone using the data corrected by the detection efficiency maps. Nor the trends, neither the exceptionality of the winter 2014/2015 (in terms of lightning counts) changed.

Minor corrections:

1. Lines 174-175. "...months (7, 9, 13, 11, 5, and 6, respectively). To create...". Please clarifies the meaning of the number between the parenthesis.

We clarified this statement on line 197 as follows: "....months (October: 7 TDs, November: 9 TDs, December: 13 TDs, January: 11 TDs, February: 5 TDs, March: 6 TDs).

2. Line 194. Please indicate how was obtained the values of 1 strokes per 3.3 km2 reported.

The number was calculated for a bin at the Norwegian coast. As there are more areas with a high concentration of lightning, we specified the text as follows on lines 216-217: "., one stroke per 3.4 km2 for the highest lightning density spot on the coast of Norway (62° N) and one stroke per 4.1 km2, for the most intense spot on the coast of Denmark (55° N).."

3. Lines 194-195. "...There was nearly no lightning activity detected above the continent." Please report the percentage of the amount of the strokes detected over the ocean and the continent.

The statement now reads on lines 218-222: "The majority of lightning activity occurred above the sea: a rough estimate based on the NOAA Global Land One-kilometer Base Elevation data (doi:10.7289/V52R3PMS) gives only 21% lighting strokes over the land. This fraction is still likely overestimated because this method tends to randomly equalize the land/sea occurrences close to the highly rugged coastlines of Northern Europe, with the WWLLN spatial uncertainty of at least several km (Rodger et al., 2005)."

4. Figure 3 b, c and d and Figure 4 b, c and d. Please considerer to use the same vertical axis in the graphics.

In the Fig. 3, we now use the same vertical axis for all the panels.

New panels c and d in Fig.3.

In the Fig. 4d, the extent of values on the vertical axes is 6 times lower than the extent of values in panels b and c. Thus, we unified only the vertical axes in panels b and c.